# Does cultural identity catalyze behavior? A TAM-based study on museum cultural creative products consumption in China

Changping Hu[1], Pengxiang Niu[1], Xiaoyi Chen[1], Pengxiang Cheng[2], Xiaojun Jiang[3], Rongfa Li[4], Ruiying Kuang 📵[5]*

1 Art Institute, Xiangtan University, Xiangtan, China, 2 Xi'an Academy of Drama, Xi'an, China, 3 School of Public Administration, Xiangtan University, Xiangtan, China, 4 Jiangxi Institute of Fashion Technology, Nanchang, China, 5 College of Art and Design, Changsha Institute of Technology, Changsha, China

* kuangruiying@xtu.edu.cn

## Abstract

China's cultural creative products are highly homogenized and can no longer meet the growing aesthetic needs of Chinese consumers. In the museum sector, the sluggish sales of cultural and creative products have become a pain point in its development. Research on consumers' perceptions and purchase intentions regarding museum cultural creative products has grown significantly in recent years, emerging as a key trend in museum sustainability and communication effectiveness studies. Existing research predominantly adopts a design perspective, evaluating designs based on aesthetic and functional needs, while neglecting the influence of consumers' cultural identity. This study investigates the influence of cultural identity on Chinese consumers' willingness to purchase museum cultural products. To achieve this, we developed a Cultural Identity Model (CIM) based on the Technology Acceptance Model (TAM). Based on the analysis of data from 404 respondents, Chinese consumers do not perceive cultural identity (P = 0.476), ethnic identity (P = 0.482), or religious identity (P = 0.242) associated with museum cultural creative products as influential factors in their purchase decisions. However, they focus on the product attributes of cultural and creative designs (materials, colors, P = 0.042) as well as functional attributes (practicality, usability and durability, P < 0.01), all of which effectively enhance consumers' purchase intention. Grounded in the cultural identity perspective, this study investigates determinants that actively catalyze purchase behavior toward museum-based cultural and creative products. The findings deliver novel design frameworks and theoretical foundations for the cultural and creative products development. This study provides empirical support for optimizing the design of museum cultural and creative products, and suggests focusing on the integration of functional attributes (such as practicality) and carriers of perceived cultural symbols (such as materials and colors).

**Data availability statement:** All relevant data are within the paper and its Supporting Information files.

**Funding:** The research project was conducted under the supervision of our academic board, and this project was supported by the foundation of Hunan Provincial Philosophy and Social Science Foundation (No.: 24ZDBQ008 to C. H.), Regional Joint Project of Hunan Provincial Natural Science Foundation (No.: 2025JJ70079 to C. H.) , Hunan Provincial Department of Education key project (No.: 24A0781 to R. K.), the Key Project of Teaching Reform Research in Hunan Provincial Education Department (No.: 202502002154 to R. K.), Hunan Provincial Department of Education Outstanding Youth Project (No.: 25B0144 to C. H.), and Sichuan University Philosophy and Social Sciences Key Research Base Industrial Design Industry Research Center Project (GYSJ2025-25 to C. H.). These funders had no role in study design, data collection and analysis, decision to publish, or preparation of the manuscript.

**Competing interests:** The authors have declared that no competing interests exist.

## Introduction

Museums have served as pivotal institutions in preserving cultural heritage, advancing public education, stimulating intellectual engagement, and fostering spiritual enrichment across civilizations [1]. In the UK, the museum industry, categorized under 'arts and culture', is a vital cultural force that enhances quality of life, education, and economic development [2]. Similarly, in China, visiting museums is considered an important way to enhance the quality of spiritual and cultural life. The prevalent linear exhibition format relies on static chronological displays, yet it struggles to foster enduring cultural memory or meaningful visitor experiences. This limitation calls for a reevaluation of institutional strategies in cultural transmission and identity building [3]. Compounding these challenges, fiscal decentralization policies and administrative restructuring within China's cultural governance framework have precipitated operational sustainability crises across museum ecosystems.

In response, curatorial innovators are pioneering heritage valorization strategies through tangible and intangible cultural asset recombination, strategically developing culturally-grounded creative commodities to stimulate consumption impulses [4]. This strategic shift aims to mitigate cultural cognition disparities between institutions and publics, and enhance participatory engagement through cultural ownership experiences. However, given that the majority of Chinese museums operate as state-owned non-profit public institutions, their cultural and creative product development inherently de-emphasizes economic value generation. This institutional orientation has precipitated significant product design homogenization, undermining consumers' capacity to establish tripartite recognition, including encompassing creative distinctiveness, aesthetic resonance, and price-value alignment. Thereby constraining purchase intention enhancement [5]. Through systematic analysis of multidimensional consumer recognition patterns toward museum cultural creative products.

This study investigates the determinants of consumer purchase intention toward museum cultural creative products in China, situated at the intersection of cultural economics, consumer behavior, and museology. And this study examines the impact of consumer cultural identity on purchase intention toward museum cultural and creative products. Through a systematic analysis of consumer cognitive mechanisms, it identifies key determinants of purchasing behavior and proposes strategic interventions to enhance consumer engagement. The findings provide theoretical and practical implications for the sustainable development of China's museum cultural creative industry. The research is motivated by the observed homogenization of Chinese cultural and creative products, which fail to meet evolving consumer aesthetic and functional demands [6]. Despite museums' role as custodians of cultural heritage [1], their museum cultural creative products often lack innovation, leading to sluggish sales, and it is a critical pain point for institutional sustainability [5]. This paper proceeds as follows: the status of museum cultural creative products section reviews literature and develops hypotheses; Methodology and objectives section describes the methodology, including sample selection and data analysis; and the discussion section discusses findings in relation to existing literature; and the conclusions and limitations section concludes with implications and limitations.

## Status of museum cultural creative products

Current scholarship on museum cultural creative products remains predominantly focused on surface-level attributes including visual aesthetics and stylistic configurations [6]. Scholars advocate that museums strategically identify collection artifacts and imagery possessing both public recognition and alignment with contemporary aesthetic preferences, subsequently distilling culturally resonant motifs for design incorporation [7]. Parallel approaches involve digitally deconstructing collection artifacts to extract culturally significant formal, contextual, and procedural characteristics that correspond to modern design paradigms [8]. These methodologies facilitate the transmission of institutional historical narratives and cultural legacies. The Metropolitan Museum of Art exemplifies this transition, having shifted from conventional merchandise (e.g., books and postcards) to culturally-embedded product lines, establishing a benchmark for commercial success [9]. This model has been widely emulated by museums globally, prompting scholars to increasingly focus on the development and design of museum cultural creative products.

Some scholars believe that the key to the success of cultural and creative products lies in consumers and the market [10]. Research on factors influencing consumers' purchasing intentions primarily focuses on the following aspects. First, research on user characteristics aims to precisely identify consumer groups, study emotional communication and resonance [11], and enhance consumers' willingness to purchase [12]. Second, focusing on functional value emphasizes the practicality of products in various usage scenarios [13]. Third, the aesthetic value of products [14], including design methods and creative thinking, is a major focus for scholars. Fourth, an increasing number of scholars highlight the educational potential of museums [15], suggesting that cultural and creative products can promote consumers' cultural identity [16,17]. Cultural identity is a multi-layered structure consisting of group-level and individual-level components, with significant differences between intergroup and intragroup cultural identities [18]. Scholars believe that museum cultural creative products can create collective social memory by encapsulating the cultural cognition of a nation or ethnic group [19]. Developing cultural and creative products using these cultural symbols can enhance consumers' cultural knowledge [20].

Cultural education serves as a catalyst for eliciting consumer affective resonance, enabling precise synchronization with consumption motivations [21]. Empirical studies establish cultural identity as a factor positively correlating with consumer purchase intention toward cultural creative products [22,23]. Threshold analysis reveals that when cultural product affinity surpasses critical thresholds, museum cultural creative products experience substantial purchase intention amplification [24]. Given China's institutional emphasis on national-ethnic identity cultivation, this investigation probes the translational potential of sociocultural belongingness in stimulating domestic museum product consumption. Concurrently, the study examines the viability of identity-informed design frameworks as sustainable developmental trajectories within museological contexts.

## Methodology and objectives

Current academic inquiry demonstrates a paucity of research examining cultural identity's impact on museum cultural creative product consumption intention. To broaden conceptual frameworks, this study establishes its theoretical foundation in the Technology Acceptance Model (TAM). The Technology Acceptance Model is frequently employed to investigate and measure audience attitudes towards adopting new technologies and their behavioral intentions to use them. An increasing number of scholars are applying TAM within research on technology acceptance in fields such as computing [25], artificial intelligence [26], and autonomous driving technology [27]. Its application is progressively expanding to encompass research on audience acceptance of novel phenomena at the psychological and cultural dimensions [28,29]. TAM's constructs of Perceived Usefulness (PU) and Perceived Ease of Use (PEOU) are adapted here to museum cultural creative products by redefining 'usefulness'as functional utility (e.g., practicality, durability) and 'ease of use' as the ease of integrating products into daily life (e.g., portability, compatibility with lifestyle). This adaptation aligns with prior research (e.g., Venkatesh et al., 2016), which demonstrates TAM's flexibility in non-technological contexts by focusing on perceived benefits and barriers to adoption. Building upon established research experiences, this methodology can

be effectively utilized to explore the impact of factors such as cultural significance, functionality, and aesthetic appeal on consumer behavior within the context of museum cultural creative products. Through methodologically adapting core TAM constructs, the investigation systematically examines the interplay between cultural identity dimensions and consumption intention metrics among Chinese museum cultural creative product consumers.

This study pursues dual objectives: (1) Systematically evaluate the salience of cultural identity among Chinese consumers regarding museum cultural creative products, and subsequently develop an optimized cultural identity framework; and (2) empirically identifying determinants of cultural receptivity through validated survey instruments, while investigating the mechanisms by which cultural creative products optimally fulfill public psycho-cultural needs.

## Theoretical model and research hypothesis

**Original theoretical TAM model.** This investigation employs the Technology Acceptance Model as its analytical framework (Fig 1) to examine Chinese consumers' perceptions of museum cultural creative products. Originating from Davis's [30] synthesis of the Theory of Reasoned Action (TRA) and Theory of Planned Behavior (TPB), the foundational TAM architecture identifies perceived usefulness (PU) and perceived ease of use (PEOU) as principal determinants [14]. Through methodological refinements in 1993 and 1996, Davis augmented the original TAM with four core constructs: perceived usefulness (PU), perceived ease of use (PEOU), attitude (ATT), and behavioral intention (BI), catalyzing extensive academic engagement. Subsequent scholarly developments include Venkatesh's TAM2 (2000), Unified Theory of Acceptance and Use of Technology [UTAUT, 2003], TAM3 (2008), and UTAUT2 (2012), alongside Lin's (2007) Technology Readiness and Acceptance Model (TRAM), collectively demonstrating TAM's theoretical adaptability across research contexts. The model's empirical validation through extensive scholarly scrutiny has solidified its position as a preeminent theoretical framework in technology adoption research, demonstrating exceptional predictive validity across information systems studies. While originally conceptualized as a predictive framework for emerging technology adoption, TAM's methodological transferability retains analytical cogency when applied to cultural creative product acceptance mechanisms. This study proposes that consumers evaluate cultural creative products, defined as artifacts incorporating tangible cultural signifiers into consumption contexts, through a dual-aspect cognitive process. This process involves simultaneous assessment of functional utility (product-related attributes) and cultural relevance (symbolic attributes). Drawing on technology acceptance theory, we identify perceived usefulness (PU) and perceived ease of use (PEOU) as antecedent determinants that systematically influence purchase decision-making mechanisms.

## Research hypothesis

**Research model framework.** Scholarly practice demonstrates systematic methodological extensions of the Technology Acceptance Model across disciplines to address context-specific technology adoption phenomena. Musa et al. found that the technology acceptance model established itself as an important research model in market research from 2002 to 2022 [31]. An and colleagues extended the Technology Acceptance Model to examine consumer adoption intentions for mobile food delivery apps [32]. Yang et al. introduced self-construal theory as cultural mediators

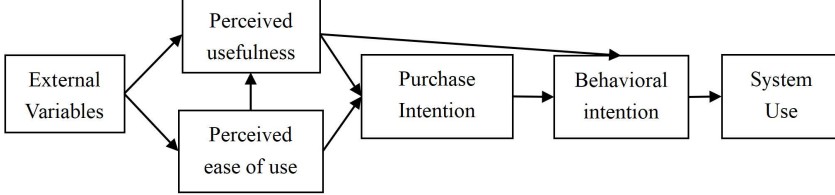

**Fig 1. Original TAM model.**

between individualism-collectivism orientations and mobile technology adoption patterns, elucidating how cultural cognition modulates TAM pathways [33]. Most recently, Zeng et al. formulated the Modified Smart Car TAM (MSCTAM) to decode aging populations' intelligent vehicle adoption barriers in China [34]. Many scholars also employ TAM to explore users' attitudes toward the use of artificial intelligence in tourism, consumption, education, and other fields [35,36,26,37,38,39,40]. These cross-disciplinary applications affirm TAM's applicability across the cultural-technological research domain.

This investigation focuses on museum cultural creative products as its research focus, diverging from conventional TAM applications predominantly centered on emerging technologies. While the foundational TAM framework proves inadequate in explicating consumption dynamics of cultural artifacts, strategic incorporation of context-specific psychosocial variables enables theoretical adaptation. Accordingly, this study proposes the Cultural Identity Model (CIM), an extension of the TAM that integrates museological contextualization and culturally adapted consumer behavior parameters (Fig 2). The CIM provides a systematic framework for analyzing purchase intention formation mechanisms. Table 1 methodically presents the model's developmental logic and theoretical innovation pathways.

While the research focus transitions from technology adoption paradigms to cultural product valuation, paradigmatic continuity persists with the original model's epistemological framework. This study examines consumer evaluation of museum cultural creative products, which simultaneously embody cultural symbolism and utilitarian functionality. Consumers assess these products through three key dimensions: perceived usefulness (PU), perceived ease of use (PEOU), and cultural signification decoding. Building on Rubera et al.'s [48] empirical findings, we posit that the resulting cultural valuation serves as the critical mediator in purchase intention formation mechanisms.

Building upon the original Technology Acceptance Model framework, this research methodologically synthesizes perceived usefulness (PU) and perceived ease of use (PEOU). The operational variables are conceptualized as functional value dimensions comprising utility, usability, and longevity [44]. Beyond functional attributes, consumer valuation of cultural creative products' symbolic properties emerges as the principal mediator governing consumer-culture interactions. It means a mechanistic parallelism to perceived ease of use (PEOU)'s role in conventional TAM. Consequently, perceived

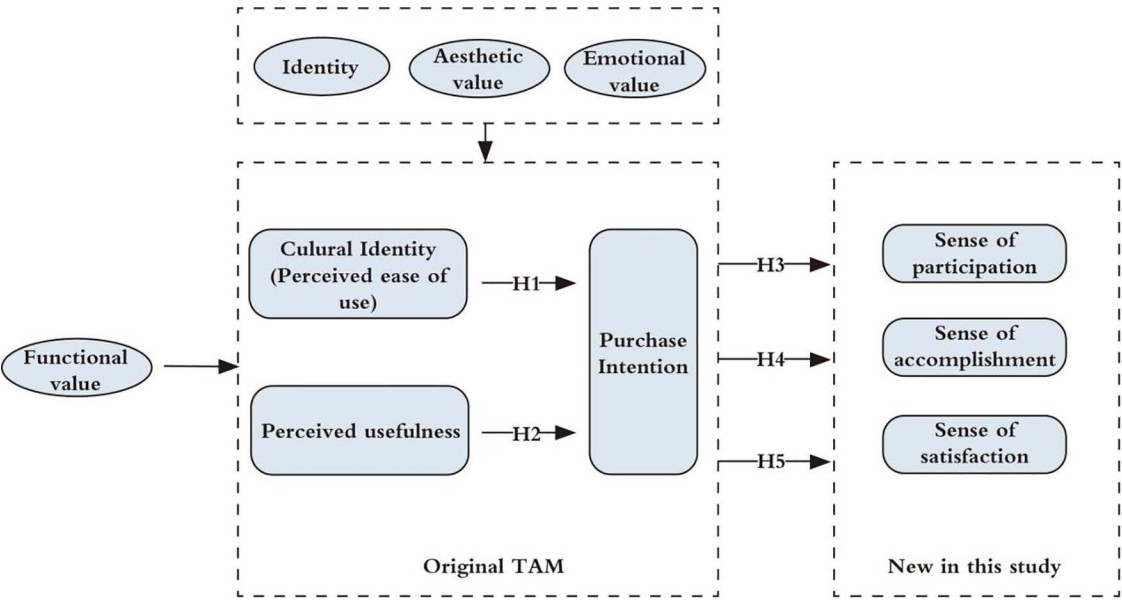

**Fig 2. Cultural identity model (CIM).**

**Table 1. TAM model development and variables.**

| Distinctions | Content | Literature |
|---|---|---|
| TAM original model | Perceived usefulness, perceived ease of use, and attitude toward use,Behavioral Intentions | [30] |
| Multi-class extended model | TAM2 (2000), UTAUT (2003), TRAM (2007), TAM3 (2008), UTAUT2 (2012) | Venkatesh, Davis, 2000; Venkatesh, Morris, 2003; Chien-Hsin Lin, 2007; Venkatesh, Bala, 2008; [41] |
| Innovative model for this study (primary) | Cultural identity: identity, aesthetic value, emotional value;<br>Perceived ease of use, functional value; purchase intention | [42,43,33];<br>[44];<br>[44] |
| Innovative model of this study (final) | Cultural Identity: aesthetic value, identity, emotional value;<br>Perceived usefulness: functional value;<br>Willingness to buy;<br>Sense of participation, access, satisfaction | [45,46];<br>[47] |

ease of use (PEOU) construct is methodologically transposed to cultural recognition dynamics, with tripartite operationalization through identity salience [43], aesthetic valuation [49], and affective resonance [33].

Methodologically accounting for the ontological distinctions between cultural creative product valuation and conventional technology adoption paradigms, this investigation introduces three novel constructs: participatory engagement [45], cultural ownership cognition [46], and experiential fulfillment [47]. These parameters systematically operationalize the transformative capacity of consumption behaviors in elevating consumers' psycho-cultural enrichment trajectories.

**Perceived usefulness and cultural recognition.** Perceived usefulness (PU) constitutes consumers' capacity to differentiate the informational utility embedded within products or services [19]. Regarding museum cultural creative products, consumers systematically evaluate functional merit through durability benchmarks, practical applicability, and usability metrics. These evaluations directly influence purchase intention formation processes. Cultural recognition manifests as consumers' perceived congruence with products' identity salience, aesthetic resonance, and emotional valence. Deficiencies in such recognition may impede the crystallization of consumer purchase intention formation mechanisms.

H1: There is a significant positive correlation between Cultural Recognition (CI) and Purchase Intention (PI).

H2: There is a significant positive correlation between Perceived Usefulness (PU) and Purchase Intention (PI).

**Identity.** Consumer selection of specific cultural creative products typically signifies endorsement of the product's encoded cultural capital, manifesting cognitive-affective alignment with its symbolic signification [20]. Museum collections inherently embody anthropological attributes spanning ethnic narratives, religious iconography, and historical consciousness. Such institutional holdings demonstrate cultural polyphony that functions as a curatorial apparatus for national identity formation [21], potentially reconfiguring visitor-artifact relational dynamics. As terminal interaction nodes within museological ecosystems, cultural creative products serve instrumental roles in forging identity salience and institutional trustworthiness, and transforming sociocultural praxis paradigms [22]. These premises yield the following propositions:

H1a: Identity Recognition (ID) is positively correlated with consumers' Purchase Intention (PI).

**Aesthetic value.** Amidst progressive enhancements in living standards, consumer aesthetic paradigms undergo continuous evolution and intensification. Museum cultural creative products systematically integrate symbolic cultural signifiers, including regional chromatic codes, historical motifs, and national identity markers, within their design lexicons [24]. Through methodological applications of deconstruction, recombination, and symmetrical patterning, designers

achieve polyvalent visual narratives [14]. Within this creative matrix, aesthetic syntax, material semiotics, structural articulation, and chromatic discourse emerge as critical determinants of design efficacy ([42],Wang, 2020). These operational principles necessitate the following proposition:

H1b: Aesthetic Value (AV) is positively correlated with consumers' Purchase Intention (PI).

**Emotional value.** Museum cultural creative products serve as instrumental mediators in museum experience augmentation. Holistic cultural engagement transcends mere epistemic acquisition to encompass cognitive profundity, imaginative liberation, affective resonance, and consumptive intentionality [50,51]. Consumer immersion in these products facilitates existential experience reconfiguration, a critical mechanism for cultivating constructive emotional schemata and determining product viability [52,53]. These operational dynamics necessitate the following proposition:

H1c: Sentimental Value (SV) is positively correlated with consumers' Purchase Intention (PI).

**Functional value.** While practical functionality may not constitute the cardinal attribute of cultural creative products, operational viability remains a critical determinant of consumer preference. Consumption patterns demonstrate preference hierarchies favoring utilitarian durability and user-centric design over ornamental artifacts, particularly when interfused with cultural semiotics [44]. Through methodological interrogation of cultural resources' axiological depth and symbolic elements across multimodal platforms, developers can engineer contemporary artifacts that synergistically integrate artistic conceptualization with functional pragmatism, thereby addressing heterogeneous market exigencies [54]. These operational imperatives necessitate the following proposition:

H2: Functional Value (FV) is positively correlated with consumers' Purchase Intention (PI).

**Sense of participation.** Paradoxically, while conventional museum narratives frequently demonstrate limited efficacy in user engagement [55], museum cultural creative products transcend spatial-temporal constraints, enabling consumers to extend institutional encounters and curate immersive cultural encounters that engender affective responses encompassing ownership cognition and collection intentionality [56]. Such experiential capital exerts substantive influence on consumption decision architectures [54]. Proactive engagement with museum cultural creative products through selection, appreciation, and identification rituals fosters cultural co-creation dynamics. And this behavioral mechanism potentially catalytic to psycho-cultural enrichment trajectories. These operational realities necessitate the following proposition:

H3: Consumers' Purchase Intention (PI) is positively correlated with Sense of Participation (SOP).

**Sense of accomplishment.** Museum experiences inherently satisfy consumer demand for cognitive enrichment, hedonic engagement, immersive interactivity, and aesthetic contemplation [57]. This requires museum administrators to adopt multisensory engagement strategies, offering high-quality cultural services and diverse products. Such approaches strategically boost public participation, enhance cultural outreach, and meet socio-educational goals [58]. As pivotal experiential conduits bridging materiality and cultural consciousness, museum cultural creative products simultaneously provide consumers with tactile artifact interactions and intangible psycho-cultural fulfillment, thereby catalyzing collective spiritual enrichment. These operational imperatives necessitate the following proposition:

H4: Consumers' Purchase Intention (PI) is positively correlated with Sense of Gain (SOG).

**Sense of satisfaction.** Beyond functional pragmatism, museum cultural creative products embody significant mnemonic capital, capable of evoking experiential nostalgia through museum visit recollection [59]. Consumer acquisitions constitute materialized expressions of cognitive-affective alignment with symbolic constructs spanning aesthetic ideology, emotional semiotics, and historical consciousness. Furthermore, these transactions facilitate socio-relational capital accumulation through gift economies and social curation practices [60]. As mnemonic artifacts, these products operationalize experience valorization through dual mechanisms: reinforcing episodic memory consolidation, and amplifying perceived experiential equity, thereby optimizing satisfaction coefficients. These operational dynamics necessitate the following proposition:

H5: Consumers' Purchase Intention (PI) is positively correlated with Sense of Satisfaction (SAT).

## Questionnaire design

Building upon the theoretical underpinnings and established variables from prior research on the Technology Acceptance Model, and integrating the distinctive design attributes of museum cultural creative products with user-specific needs, the questionnaire was methodologically structured. Through preliminary research, five dependent variables and 28 independent variables affecting consumers' cultural recognition of museum cultural creative products were identified (Table 2). The five dependent variables are Cultural Identity (CID), Perceived Usefulness (PU), Sense of Participation (SOP), Sense of Gain (SOG), and Sense of Satisfaction (SAT). The independent variables include:Identity Recognition (ID), which comprises Cultural Identity (CI), Ethnic Identity (EID), and Religious Identity (RID). Aesthetic Identity (AESI), which includes Form (FOM), Structure (CONS), Material (MAT), and Color (CLR).Emotional Identity (EI), which includes Emotional Experience (EEXP), Emotional Attitude (EA), and Emotional Behavior (AB).Functional Value (FVAL), which consists of Utility (UTY), Ease of Use (UF), and Durability (DURA).Sense of Participation (SOP), measured by Emotional Resonance (ER), Experiential Feeling (EXPFL), and Interactivity (IS).Sense of Gain (SOG), which includes Pleasure (PLE), Fun (ENJ), Intellectual (INTEL), Cultural Confidence (CULC), and Spiritual Richness(SR).Sense of Satisfaction (SAT), which includes Collection Satisfaction (COLLS), Aesthetic Satisfaction (AESS), and Social Satisfaction (SOCS). All items are measured using a Likert 5-point scale (1=strongly disagree; 2=disagree; 3=neutral; 4=agree; 5=strongly agree), and respondents are asked to rate each item for every construct.

**Data collection.** A randomized sampling method was employed to select Chinese residents across multiple provinces who had visited museums at least once. Given that purchasing behavior and potential demand for museum-based cultural and creative products are age-independent, participants under 18 years old were included in the survey following guardian consent, provided that their comprehension of the questionnaire was verified. The empirical data collection protocol employed dual-modality administration (digital and in-person), yielding a total pool of 438 questionnaire responses. The online portion of the questionnaire was collected and organized using the website Wenjuanxing (https://www.wjx.cn/). Participants completed the survey via mobile phones, with each IP address restricted to a single submission, thereby enhancing the reliability of the sample. The offline visits phase employs semi-structured interviews to collect responses, where data undergo systematic categorization and thematic interpretation processes to identify core themes and analytical insights. Following rigorous validation procedures, 24 submissions were excluded due to response incompleteness, resulting in 404 valid datasets with balanced gender representation (male: n=200; female: n=204) comprising the final analytical cohort.

**Descriptive statistical analysis.** From the statistical results (Table 3), the proportion of male and female respondents was 49.5% and 50.5%, respectively, indicating a relatively balanced gender ratio. Among the respondents, those aged 14–19 and 20–39 accounted for more than half of the total. Additionally, 42.08% of respondents had a bachelor's or associate degree, and 23.27% had a master's degree or higher. Moreover, 45.05% of respondents had an income level below 999 RMB/month, indicating a high proportion of students. Observations from random samples reveal that museum visitors are predominantly low-income students, consistent with Crispin and his colleagues' [61] research indicating that youth museum attendance rates typically range between 50% and 70%.

**Reliability and validity tests.** Following data purification procedures, the analytical protocol employed SPSS 27.0 for psychometric evaluation, commencing with reliability assessment via Cronbach's Alpha coefficient [62]. As a widely adopted metric in psychometric testing, Cronbach's Alpha quantifies internal consistency reliability through covariance analysis, representing the primary psychometric index in social science research. The coefficient's magnitude directly correlates with measurement precision, with established benchmarks requiring Alpha≥0.80 for confirmatory studies and Alpha≥0.70 for exploratory studies. As detailed in Table 4, the global Alpha coefficient for this instrument reached 0.750, with standardized Alpha=0.762 across all latent constructs. All subdimensional Alpha values exceeded the 0.70 threshold, demonstrating satisfactory psychometric properties across all measured constructs.

Before conducting specific factor analysis, this study also performed the KMO and Bartlett's tests to determine whether the correlations between the selected variables were sufficient. Passing these tests allows for more accurate factor

**Table 2. Detailed variable factors.**

| Construct | | Item | | Explanation | References |
|---|---|---|---|---|---|
| Cultural Identity(Perceived Ease of Use) (CID) | Identity Recognition (ID) | Cultural Identity(CI) | ID1 | The historical and cultural elements embodied in museum cultural creative products can enhance my purchase intention. | [21,33] |
| | | Ethnic Identity(EID) | ID2 | The ethnic cultural elements embodied in museum cultural creative products can enhance my purchase intention. | [22,21] |
| | | Religious Identity(RID) | ID3 | The religious cultural elements (such as Buddhism, Taoism, etc.) embodied in museum cultural creative products can enhance my purchase intention. | |
| | Aesthetic Identity (AESI) | Form(FOM) | AE1 | I care deeply about the overall appearance of museum cultural creative products. | [42,14]; Wang, 2020; [1] |
| | | Structure (CONS) | AE2 | I care deeply about the design creativity of museum cultural creative products. | |
| | | Material(MAT) | AE3 | I care deeply about the materials used in museum cultural creative products. | |
| | | Color(CLR) | AE4 | I care deeply about the aesthetic appeal of the colors in museum cultural creative products. | |
| | Emotional Identity (EI) | Emotional Experience (EEXP) | EI1 | I expect to gain emotional experiences (such as joy, sadness, pride, etc.) from museum cultural creative products. | [42] |
| | | Emotional Attitude(EA) | EI2 | I believe I will have selectable emotional attitudes such as acceptance or resistance towards museum cultural creative products. | [52] |
| | | Emotional Behavior(AB) | EI3 | I expect to engage in emotional behaviors (such as appreciation, touching, cherishing, etc.) towards museum cultural creative products. | Liu et al., 2019 |
| Perceived Usefulness(PU) | Functional Value (FVAL) | Utility(UTY) | FV1 | I believe that museum cultural creative products should be practical. | Hsueh et al., 2021; [50,44] |
| | | Ease of Use(UF) | FV2 | I believe that museum cultural creative products should be easy to use. | |
| | | Durability (DURA) | FV3 | I believe that museum cultural creative products should be durable. | |
| Sense of Participation (SOP) | | Emotional Resonance (ER) | SP1 | I hope to experience emotional resonance with museum cultural creative products, such as feelings of joy, solemnity, and depth. | [43]; Cole, 2007; [54]; [55] |
| | | Experiential Feeling (EXPEL) | SP2 | I believe that interacting with museum cultural creative products can provide me with positive experiences, such as entertainment, education, immersion, and aesthetics. | |
| | | Interactivity(IS) | SP3 | I believe I will interact with museum cultural creative products, such as by handling or appreciating them. | |
| Sense of Gain (SOG) | | Pleasure(PLE) | SG1 | I believe that museum cultural creative products can bring me a sense of pleasure. | Easson,Leask, 2019; Ren, 2022 |
| | | Pleasure(ENJ) | SG2 | I believe that museum cultural creative products should be interesting. | |
| | | Intellectual (INTEL) | SG3 | I believe that museum cultural creative products can provide me with cultural knowledge. | |
| | | Cultural Confidence (CULC) | SG4 | I believe that museum cultural creative products can instill cultural confidence in me. | |
| | | Spiritual richness (SR) | SG5 | I believe that owning museum cultural creative products is a way to enrich one's spiritual and cultural life. | |

*(Continued)*

**Table 2.** (Continued)

| Construct | Item | | Explanation | References |
|---|---|---|---|---|
| Sense of Satisfaction (SAT) | Collection Satisfaction(COLLS) | SA1 | Owning museum cultural creative products can satisfy my desire for collecting. | Easson,Leask, 2019; Ren, 2022; Ponimin et al., 2021; [60] |
| | Aesthetic Satisfaction(AESS) | SA2 | Owning museum cultural creative products can provide me with positive psychological feelings, such as a sense of achievement, confidence, and pride. | |
| | Social Satisfaction(SOCS) | SA3 | Owning museum cultural creative products positively influences my social activities, such as sharing cultural creations and exchanging cultural knowledge. | |
| Purchase Intention (PI) | | PI1 | Cultural identity (including identity recognition, aesthetic value, and emotional value) has a significant impact on my purchase intention. | [19,20]; Yoon, Cho, 2015 |
| | | PI2 | Perceived usefulness (functional value) has a significant impact on my purchase intention. | |
| | | PI3 | If I have a purchase need, I will consider cultural identity, perceived ease of use (including identity recognition, aesthetic value, emotional value, and functional value) before making a purchase. | |

**Table 3.** Descriptive statistics of the sample.

| Variable | Categorization | Proportion/% (N = 404) | Percentage(%) |
|---|---|---|---|
| **Gender** | Male | 200 | 49.5 |
| | Female | 204 | 50.5 |
| **Age** | 4-13 years | 37 | 9.16 |
| | 14-19 years | 118 | 29.21 |
| | 20-39 years | 206 | 50.99 |
| | 40-59 years | 20 | 4.95 |
| | 60 and over | 23 | 5.69 |
| **Education level** | Junior high school and below | 68 | 16.83 |
| | High school or junior college | 72 | 17.82 |
| | Bachelor's or associate degree | 170 | 42.08 |
| | Master's degree or higher | 94 | 23.27 |
| **Monthly salary** | 999 RMB and below | 182 | 45.05 |
| | 1000-2999 RMB | 41 | 10.15 |
| | 3000-5999 RMB | 104 | 25.74 |
| | 6000-9999 RMB | 51 | 12.62 |
| | 10,000RMB and above | 26 | 6.44 |

analysis and reliable results. When the KMO value is greater than 0.6, it indicates that factor analysis can be performed. As shown in Table 5, the KMO value of this survey questionnaire is greater than 0.8, and the significance value is 0.000, indicating that the data is suitable for further analysis.

## Results

This study employed AMOS 23.0 to construct the theoretical model and import the questionnaire data. The model was estimated using the Maximum Likelihood (ML) estimation method, and the resulting path coefficients and fit indices are presented in Fig 3 and Table 6, respectively. The model fit indices, including $\chi^2 = 3.390$, RMSEA = 0.077, GFI = 0.796,

**Table 4. Validity analysis.**

| Cronbach Alpha | Normalized term-based clone Bach Alpha | item count (of a consignment etc) |
|---|---|---|
| .750 | .762 | 27 |

**Table 5. KMO and Bartlett's test.**

| KMO Number of Sampling Suitability Measure. | | .818 |
|---|---|---|
| Bartlett's test of sphericity | Approximate chi-square | 2899.644 |
| | degrees of freedom | 351 |
| | significance | .000 |

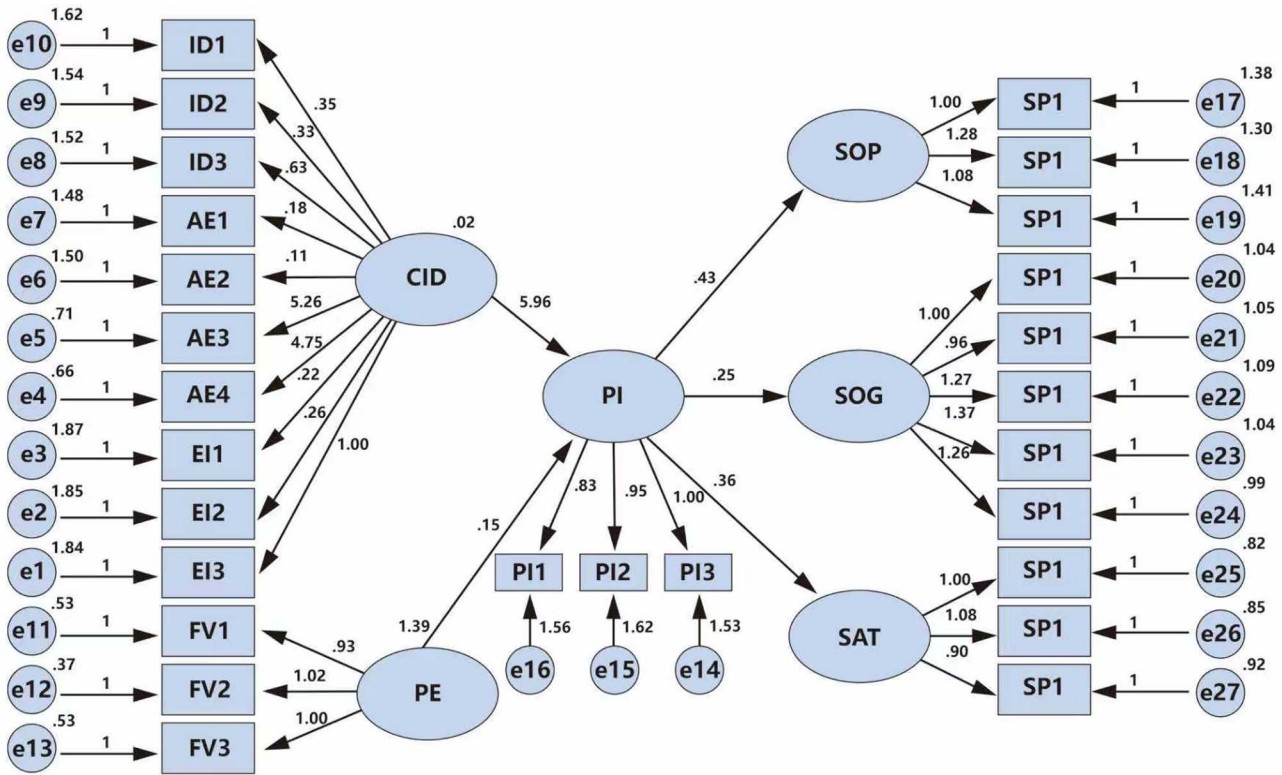

**Fig 3. Results of the model test.**

AGFI = 0.761, NFI = 0.705, IFI = 0.581, and CFI = 0.708, indicate an acceptable model fit according to commonly accepted thresholds (e.g., RMSEA < 0.08, CFI > 0.70). In Table 6, which shows the model path coefficients, the p-values for hypotheses H1, H2, H3, H4, and H5 are all less than 0.05, indicating significant positive effects. Therefore, the Cultural Identity Model (CIM), an innovative extension of the Technology Acceptance Model, fits the data well (Table 7).

## Discussion

Contrary to theoretical postulations positing sociocultural identity constructs as determinants of sustainable consumption patterns [63], empirical findings negate support for hypothesis H1. The sociocultural identity constructs (cultural [CI],

**Table 6. Model fitting results.**

| Cardinality/degrees of freedom | RMSEA | GFI | AGFI | NFI | CFI | IFI |
|---|---|---|---|---|---|---|
| <5 | <0.08 | >0.7 | >0.7 | >0.7 | >0.7 | >0.5 |
| 1094.931/323 = 3.390 | 0.077 | 0.796 | 0.761 | 0.705 | 0.708 | 0.581 |

**Table 7. Table of model path coefficients.**

| Hypothesis | Hypothesised path | | | Estimate | S.E. | C.R. | P | Supported |
|---|---|---|---|---|---|---|---|---|
| H1 | PI | ← | PU | 0.145 | 0.031 | 4.631 | *** | YES |
| H2 | PI | ← | CID | 5.962 | 2.927 | 2.037 | 0.042 | YES |
| H3 | SOG | ← | PI | 0.249 | 0.061 | 4.063 | *** | YES |
| H4 | SAT | ← | PI | 0.364 | 0.056 | 6.505 | *** | YES |
| H5 | SOP | ← | PI | 0.426 | 0.072 | 5.908 | *** | YES |
| | EI3 | ← | CID | 1 | | | | |
| | EI2 | ← | CID | −0.263 | 0.506 | −0.519 | 0.603 | No |
| | EI1 | ← | CID | 0.218 | 0.503 | 0.434 | 0.664 | No |
| | AE4 | ← | CID | 4.747 | 2.34 | 2.029 | 0.042 | YES |
| | AE3 | ← | CID | 5.262 | 2.591 | 2.031 | 0.042 | YES |
| | AE2 | ← | CID | 0.111 | 0.444 | 0.249 | 0.803 | No |
| | AE1 | ← | CID | 0.183 | 0.448 | 0.409 | 0.683 | No |
| | ID3 | ← | CID | 0.633 | 0.541 | 1.171 | 0.242 | No |
| | ID2 | ← | CID | −0.334 | 0.476 | −0.702 | 0.482 | No |
| | ID1 | ← | CID | 0.349 | 0.489 | 0.713 | 0.476 | No |
| | FV3 | ← | PU | 1 | | | | |
| | FV2 | ← | PU | 1.023 | 0.049 | 20.995 | *** | YES |
| | FV1 | ← | PU | 0.934 | 0.047 | 19.777 | *** | YES |
| | PI3 | ← | PI | 1 | | | | |
| | PI2 | ← | PI | 0.952 | 0.065 | 14.734 | *** | YES |
| | PI1 | ← | PI | 0.83 | 0.059 | 14.051 | *** | YES |
| | SP1 | ← | SOP | 1 | | | | |
| | SP2 | ← | SOP | 1.277 | 0.262 | 4.877 | *** | YES |
| | SP3 | ← | SOP | 1.081 | 0.242 | 4.478 | *** | YES |
| | SG1 | ← | SOG | 1 | | | | |
| | SG2 | ← | SOG | 0.963 | 0.337 | 2.858 | 0.004 | YES |
| | SG3 | ← | SOG | 1.274 | 0.395 | 3.223 | 0.001 | YES |
| | SG4 | ← | SOG | 1.375 | 0.412 | 3.34 | *** | YES |
| | SG5 | ← | SOG | 1.202 | 0.374 | 3.21 | 0.001 | YES |
| | SA1 | ← | SAT | 1 | | | | |
| | SA2 | ← | SAT | 1.083 | 0.219 | 4.949 | *** | YES |
| | SA3 | ← | SAT | 0.897 | 0.204 | 4.396 | *** | YES |

***indicates significant at the 0.01 level.

ethnic [EID], and religious [RID] dimensions) demonstrated non-significant associations with purchase intention (PI), yielding p-values of 0.476 (CI→PI), 0.482 (EID→PI), and 0.242 (RID→PI). All exceeding the Alpha = 0.05 significance threshold. This result substantiates that Chinese consumers' conscious recognition of museological products' cultural, ethnic,

and religious semiotics remains non-predictive of consumption decisions. Crucially, even when perceptually decoded, these elements fail to attain determinative salience in consumption decision matrices.

Hypothesis H2 testing revealed differential effects within aesthetic identity (AESI) dimensions. While form (FOM: $p = 0.683$) and structure (CONS: $p = 0.803$) demonstrated non-significant associations with purchase intention (PI), exceeding the Alpha = 0.05 statistical significance threshold, material (MAT: $p = 0.042$) and color (CLR: $p = 0.042$) components exhibited statistically significant positive correlations. This empirical finding corroborates Wu's (2021) assertion regarding the positive influence of color on consumer purchasing behavior. It also supports the conclusion drawn by Huang et al. [42] that among various cultural attributes of cultural and creative products, consumers place the highest importance on "the materials used." The findings substantiate market preferences for cultural creative products emphasizing instantaneous perceptual accessibility through sensory optimization rather than elaborate formal configurations.

The emotional identity (EI) framework manifests differential predictive efficacy across its constituent dimensions. Emotional experience (EEXP: $p = 0.664$) and emotional attitude (EA: $p = 0.603$) demonstrated non-significant correlations with purchase intention (PI), surpassing the Alpha = 0.05 significance threshold, while emotional behavior (AB: $p < 0.05$) attained statistical significance. This finding supports Rodriguez et al. [52]'s assertion on the significance of emotional value in museums and further elucidates the positive impact of emotional behavior on the purchase intention for museum cultural creative products. This difference stems from emotion-based processes influenced by time. Long-term emotional accumulation and passive experience-based consumption have reduced ability to trigger impulse buying. In contrast, interactive behaviors involving touch, visual assessment, and memory stimulation can revive dormant emotions from museum experiences. And this ultimately increases purchasing tendency through immersive re-experiencing mechanisms.

Hypothesis H2 proposes a statistically significant positive correlation between perceived usefulness (PU) and purchase intention (PI) ($p < 0.05$). This relationship measures PU through functional value dimensions (FVAL). The results confirm the hypothesis H2. This empirical alignment likely derives from consumers' prioritization of tripartite evaluative criteria: utility (UTY) optimization, usability (UF) efficiency, and durability (DURA) assurance when engaging with cultural creative products. Building on the theoretical foundation established by Moslehpour et al. [19] regarding the sustained positive effects of perceived usefulness and ease of use on consumer purchase intention, this empirical study corroborates the applicability of the Technology Acceptance Model in the museum-based cultural and creative products domain, particularly through its extended integration with symbolic consumption theory.

Hypothesis H3 substantiates that Chinese consumers' acquisition of cultural creative products significantly enhances perceived cultural participation ($p < 0.05$). Post-purchase engagement facilitates tripartite experiential outcomes: affective resonance (ER), experiential fulfillment (EXPFL), and interactive symbiosis (IS). This mechanism shows how consumer-product interactions reactivate emotional connections to museums beyond their physical and time constraints. These interactions occur through practical use, playful engagement, and aesthetic appreciation. Consequently, they amplify participation in cultural narratives. This enhanced participation happens through repeated experiences outside museum settings. This form of cultural participation demonstrates alignment with consumer preferences, which has been empirically validated to positively enhance cultural identity formation [43].

Hypothesis H4 suggests that purchasing cultural and creative products can give consumers a sense of gain (SOG) in museum culture ($p < 0.05$), supporting this hypothesis. To some extent, this reflects that consumers can reinforce and solidify their museum cultural experiences and memories through pleasure (PLE), enjoyment (ENJ), knowledge (INTEL), cultural confidence (CULC), and spiritual enrichment (SR). This also means that Chinese people can gain spiritual and cultural experiences by purchasing museum cultural creative products [63], the consumer behavior can enhance cultural acquisition and contribute to the formation of cultural identity to some extent.

Hypothesis H5 states that purchasing cultural and creative products can enhance consumers' satisfaction with cultural needs ($p < 0.05$). This hypothesis substantiates Yan's (2019) research findings, demonstrating that the processes of

collecting, appreciating, owning, and sharing purchased museum-based cultural and creative products facilitate meeting consumers' expectations for spiritual-cultural enrichment.

Summary of Supported Hypotheses (Table 8).

The finding of the research shows that functional and aesthetic attributes outweigh cultural identity. Empirical findings demonstrate Chinese consumers maintain an optimistic disposition toward museum cultural creative product acquisition, characterized by anticipatory openness, albeit moderated by contextual variables. Respondents perceive these artifacts as fulfilling dual imperatives: spiritual enrichment, lifestyle elevation, and encompassing functional materiality and aesthetic-cultural aspirations. This research found that the non-significance of cultural identity may reflect China's museum cultural creative product market maturity: as products have become more commercialized, consumers prioritize utility over symbolism. This study differs from previous studies [22,23] that concluded that cultural identity can positively influence purchase intention. Additionally, younger respondents may associate museum cultural creative products with trendy aesthetics rather than cultural heritage, aligning with global youth consumer trends.

While, Chinese museum cultural creative products exhibit homogeneous strategic positioning that prioritizes cultural connotations while neglecting product utility. Through intensifying consumers' cultural identity, institutions attempt to elevate purchase intention toward upscale products. Market offerings, engineered to mirror institutional prestige through structural complexity and avant-garde formal aesthetics, adopt premium pricing strategies. Paradoxically, China's museum visitor demographics are also primarily composed of low-income youth populations with limited purchasing power. Especially student visitors possess cultural decoding competence but face budget constraints, so they rely on readily evaluable cues (material, color, utility) rather than symbolic value. In the current Chinese museum cultural creative products market context, functional and sensory attributes exert a stronger immediate influence on purchase decisions; cultural narratives should be embedded through perceivable channels such as material and color rather than relying solely on symbolic labels. This orientation focuses disproportionately on premium consumption segments yet inadequately addresses mass cultural demands. Such approaches neither align with museums' social function as public cultural entities, nor acknowledge the catalytic role of inherent product attributes in driving purchasing behavior [64]. Consequently, product development should underserved consumer groups by diversifying product features, refining development processes, reducing

**Table 8. Variable Factors.**

| Cultural Identity (Perceived Ease of Use) (CID) | Aesthetic identity(AESI) | Material(MAT) |
| --- | --- | --- |
| | | Color(CLR) |
| Perceived Usefulness(PU) | Functional value(FVAL) | Utility(UTY) |
| | | Ease of Use(UF) |
| | | Durability (DURA) |
| Sense of Participation(SOP) | | Emotional Resonance (ER) |
| | | Experiential Feeling (EXPEL) |
| | | Interactivity(IS) |
| Sense of Gain (SOG) | | Pleasure(PLE) |
| | | Pleasure(ENJ) |
| | | Intellectual (INTEL) |
| | | Cultural Confidence (CULC) |
| | | Spiritual richness (SR) |
| Sense of Satisfaction(SAT) | | Collection Satisfaction(COLLS) |
| | | Aesthetic Satisfaction(AESS) |
| | | Social Satisfaction(SOCS) |

consumption barriers, and enhancing market accessibility. This product-consumer socioeconomic incongruence constitutes a critical market penetration barrier, potentially compromising museological cultural dissemination efficacy.

The institutional paradigm of Chinese museums as public service entities inherently limits commercial promotion of cultural creative products. This operational constraint diminishes product visibility within visitor engagement cycles, necessitating strategic marketing campaigns to elevate consumer awareness and purchasing intentionality. In the sustainable development strategy, Cultural institutions should give full play to their functions of social culture, economic exchange and environmental sustainability [65]. Museums should leverage alternative educational channels to enhance cultural literacy and cultivate demand among current and potential consumers, thereby establishing circular consumption models to activate latent market potential. Concurrently, design protocols must preserve cultural veracity while integrating user-centric design methodologies to engineer intuitive, functional, and economically viable quotidian commodities aligned with consumer anticipations.

## Conclusions and limitations

### Conclusions

Questionnaire-based research is instrumental in systematically capturing audiences' tangible needs. These needs include affective responses, aesthetic expectations, cultural resonance, and ergonomic experiences. This approach provides empirically grounded evidence for implementing design strategies [66]. To examine the consumption demand for museum-based cultural and creative products in China, this study explores determinants of consumers' purchase intention toward these products, building upon the foundational framework of the Technology Acceptance Model. Through a survey of 404 Chinese residents, the study finds that perceived ease of use and perceived usefulness significantly affect Chinese users' purchase intentions for cultural and creative products. Moreover, purchasing cultural and creative products influences users' sense of cultural participation, cultural gain, and cultural satisfaction.

The results indicate that materials and colors in aesthetic identity, emotional behavior in emotional identity, and functional value in perceived usefulness significantly impact users' purchase intentions for museum cultural creative products. The findings reveal that cultural identity exerts limited influence on purchase behavior toward museum-based cultural and creative products. Consequently, museums and relevant cultural institutions should prioritize product attributes during the design and planning phases, rather than overemphasizing cultural value enhancement.

In summary, our work contributes to optimizing the design methods of museum cultural creative products and emphasizes the importance of aesthetic identity, emotional identity, functional value, participation, gain, and satisfaction in the design of these products. We hope these findings will expand our understanding of users' emotional, value, and functional needs, thereby promoting the sustainable development of museums.

### Limitations and future research directions

While this investigation advances scholarly understanding of Chinese consumers' perceptual frameworks toward museum cultural creative products across cultural, aesthetic, and functional dimensions, several methodological constraints merit acknowledgement. We also acknowledge potential limitations, such as the measure of cultural identity and suggest future research use multi-item scales for sub-dimensions. The limited sample representativeness and geographical scope compromise the external validity of findings. Given the ethnoculturally homogeneous cohort, the findings possess inherent cultural specificity to Chinese consumers' evaluations of domestic museological products. Comparative cross-cultural investigations remain imperative to elucidate international consumers' interpretative frameworks regarding Chinese museum cultural creative products.

Future investigations should prioritize methodological enhancements through two strategic pathways: (1) augmenting sample representativeness via expanded cohort diversity and volume, and (2) implementing multi-domain validation protocols for the Cultural Identity Model (CIM) proposed in this study.

While the current CIM framework synthesizes global scholarly insights with comprehensive empirical foundations, subsequent research must pursue: (1)Model Refinement: Systematic enrichment through longitudinal ethnographic user engagement data collection to capture evolving consumption patterns in museological markets. The variables are detailed, and the respondent sample is biased toward certain age and income groups. In future studies, it is necessary to secure sample representativeness and simplicity in variable manipulation.it also important that the lack of examination of the mediating or moderating effects of cultural identity. (2)Cross-Contextual Verification: Empirical substantiation of the Cultural Identity Model (CIM) framework through multi-dimensional analytical matrices encompassing cross-cultural, cross-industrial, and cross-demographic perspectives. These initiatives will strengthen the model's theoretical robustness while advancing pragmatic understanding of contemporary museum cultural creative product consumption dynamics. (3)Future research could also include comparative studies across different regions of China to examine the impact of regional cultural differences on consumer behavior regarding museum cultural creative products. Furthermore, cross-cultural comparative studies between Eastern and Western countries could be conducted, using diverse cultural samples to verify the impact of the cultural attributes of museum cultural creative products on consumer behavior. This multi-level comparative study would help reveal the universality and specificity of cultural factors in the consumption of cultural and creative products.

## Supporting information

**S1 File. Questionnaire on the study of factor was uploaded on the website 'https://www.editorialmanager.com/pone'.**
(XLSX)

## Author contributions

**Conceptualization:** Changping Hu, Ruiying Kuang.

**Data curation:** Xiaoyi Chen.

**Formal analysis:** Changping Hu, Xiaoyi Chen.

**Methodology:** Changping Hu, Ruiying Kuang.

**Validation:** Changping Hu, Pengxiang Niu, Pengxiang Cheng, Xiaojun Jiang, Rongfa Li.

**Visualization:** Xiaoyi Chen.

**Writing – original draft:** Changping Hu, Xiaoyi Chen.

**Writing – review & editing:** Changping Hu, Pengxiang Niu, Ruiying Kuang.

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
