## [Decision Letter · Decision Letter 0]

14 Sep 2025

PONE-D-25-38750Does Cultural Identity Catalyze Behavior? A TAM-Based Study on Museum Cultural Creative Products Consumption in ChinaPLOS ONE?

Dear Dr. Kuang,

Thank you for submitting your manuscript to PLOS ONE. After careful consideration, we feel that it has merit but does not fully meet PLOS ONE’s publication criteria as it currently stands. Therefore, we invite you to submit a revised version of the manuscript that addresses the points raised during the review process.

**Please revise the manuscript according to the comments from reviewers.**

We look forward to receiving your revised manuscript.

Kind regards,

Qianda Zhuang, Ph.D.

Guest Editor

PLOS ONE

**Journal Requirements:**

1. When submitting your revision, we need you to address these additional requirements. Please ensure that your manuscript meets PLOS ONE's style requirements, including those for file naming. The PLOS ONE style templates can be found at https://journals.plos.org/plosone/s/file?id=wjVg/PLOSOne_formatting_sample_main_body.pdf and https://journals.plos.org/plosone/s/file?id=ba62/PLOSOne_formatting_sample_title_authors_affiliations.pdf 2. We note that the grant information you provided in the ‘Funding Information’ and ‘Financial Disclosure’ sections do not match.  When you resubmit, please ensure that you provide the correct grant numbers for the awards you received for your study in the ‘Funding Information’ section. 3. Thank you for stating in your Funding Statement: The research project was conducted under the supervision of our academic board, and this project was supported by the foundation of the Key Project of Hunan Provincial Philosophy and Social Science foundation(Project No.: 24ZDBQ008)-Changping Hu, Regional Joint Project of Hunan Provincial Natural Science Foundation(Project No.: 2025JJ70079)-Changping Hu and Hunan Provincial Department of Education key project (Project No.: 24A0781)-Ruiying Kuang.  Please provide an amended statement that declares *all* the funding or sources of support (whether external or internal to your organization) received during this study, as detailed online in our guide for authors at http://journals.plos.org/plosone/s/submit-now.  Please also include the statement “There was no additional external funding received for this study.” in your updated Funding Statement. Please include your amended Funding Statement within your cover letter. We will change the online submission form on your behalf. 4. Thank you for stating the following in the Acknowledgments Section of your manuscript: The research project was conducted under the supervision of our academic board, and this project was supported by the foundation of the Key Project of Huxiang Culture Youth Heritage Scholar Cultivation Program, Hunan Provincial Philosophy and Social Science Foundation(Project No.: 24ZDBQ008), Regional Joint Project of Hunan Provincial Natural Science Foundation(Project No.: 2025JJ70079) and Hunan Provincial Department of Education key project (Project No.: 24A0781). We note that you have provided funding information that is not currently declared in your Funding Statement. However, funding information should not appear in the Acknowledgments section or other areas of your manuscript. We will only publish funding information present in the Funding Statement section of the online submission form. Please remove any funding-related text from the manuscript and let us know how you would like to update your Funding Statement. Currently, your Funding Statement reads as follows: The research project was conducted under the supervision of our academic board, and this project was supported by the foundation of the Key Project of Hunan Provincial Philosophy and Social Science foundation(Project No.: 24ZDBQ008)-Changping Hu, Regional Joint Project of Hunan Provincial Natural Science Foundation(Project No.: 2025JJ70079)-Changping Hu and Hunan Provincial Department of Education key project (Project No.: 24A0781)-Ruiying Kuang.  Please include your amended statements within your cover letter; we will change the online submission form on your behalf. 5. In the online submission form you indicate that your data is not available for proprietary reasons and have provided a contact point for accessing this data. Please note that your current contact point is a co-author on this manuscript. According to our Data Policy, the contact point must not be an author on the manuscript and must be an institutional contact, ideally not an individual. Please revise your data statement to a non-author institutional point of contact, such as a data access or ethics committee, and send this to us via return email. Please also include contact information for the third party organization, and please include the full citation of where the data can be found. 6. When completing the data availability statement of the submission form, you indicated that you will make your data available on acceptance. We strongly recommend all authors decide on a data sharing plan before acceptance, as the process can be lengthy and hold up publication timelines. Please note that, though access restrictions are acceptable now, your entire data will need to be made freely accessible if your manuscript is accepted for publication. This policy applies to all data except where public deposition would breach compliance with the protocol approved by your research ethics board. If you are unable to adhere to our open data policy, please kindly revise your statement to explain your reasoning and we will seek the editor's input on an exemption. Please be assured that, once you have provided your new statement, the assessment of your exemption will not hold up the peer review process. 7. PLOS requires an ORCID iD for the corresponding author in Editorial Manager on papers submitted after December 6th, 2016. Please ensure that you have an ORCID iD and that it is validated in Editorial Manager. To do this, go to ‘Update my Information’ (in the upper left-hand corner of the main menu), and click on the Fetch/Validate link next to the ORCID field. This will take you to the ORCID site and allow you to create a new iD or authenticate a pre-existing iD in Editorial Manager. 8. Please include your full ethics statement in the ‘Methods’ section of your manuscript file. In your statement, please include the full name of the IRB or ethics committee who approved or waived your study, as well as whether or not you obtained informed written or verbal consent. If consent was waived for your study, please include this information in your statement as well. 9. If the reviewer comments include a recommendation to cite specific previously published works, please review and evaluate these publications to determine whether they are relevant and should be cited. There is no requirement to cite these works unless the editor has indicated otherwise. 

Reviewers' comments:

**Comments to the Author**

1. Is the manuscript technically sound, and do the data support the conclusions?

Reviewer #1: Yes

Reviewer #2: Yes

2. Has the statistical analysis been performed appropriately and rigorously?

Reviewer #1: Yes

Reviewer #2: Yes

3. Have the authors made all data underlying the findings in their manuscript fully available?

Reviewer #1: Yes

Reviewer #2: Yes

4. Is the manuscript presented in an intelligible fashion and written in standard English?

Reviewer #1: Yes

Reviewer #2: Yes

**Reviewer #1:**  1. In the introduction, please add main sources and the field of your research, as well as the theoretical gap you intend to fill in. Besides, please introduce the structure of your paper here.

2. In the discussions, please compare your results with other results.

3. The list of references should be composed according to the alphabet.

4. I hope the following source could be helpful for you.

Kačerauskas, T. 2023. Disinterestedness in the creative economy: the case of the MO museum in Vilnius. Sustainability 15 (13): 10115.

**Reviewer #2:**  Reviewer Report

Manuscript ID: PONE-D-25-38750

Title: Does Cultural Identity Catalyze Behavior? A TAM-Based Study on Museum Cultural Creative Products Consumption in China

Journal: PLOS ONE

Decision: Minor Revision

Overall Evaluation:

This manuscript presents a well-structured and timely study investigating the drivers behind the consumption of museum cultural creative products (MCCPs) in China. The research is grounded in the well-established Technology Acceptance Model (TAM), which is innovatively extended to create a Cultural Identity Model (CIM). The topic is relevant, the methodology is generally sound (using SEM on a sample of 404 respondents), and the findings offer valuable practical insights for museum management and product designers. The key finding—that functional and aesthetic attributes outweigh cultural identity in driving purchase intention among Chinese consumers—is interesting and somewhat counter-intuitive, contributing meaningfully to the literature.

The manuscript is largely publication-ready. The revisions requested are minor and primarily concern enhancing clarity, providing further justification for certain methodological choices, and slightly deepening the interpretation of the results.

Specific Comments for Minor Revision:

Clarity and Flow:

The manuscript is generally well-written. However, some sentences, particularly in the introduction and theoretical sections, are very long and complex. I recommend breaking a few of these down for improved readability and flow.

Example: The sentence beginning "The prevalent linear exhibition format..." on Page 10 is quite dense.

Theoretical Justification:

The adaptation of TAM (a model for technology adoption) to the context of cultural products is the core innovation of this paper. While the authors do explain this, a more robust justification in the Methodology and objectives section would be beneficial. Briefly elaborate on why the constructs of PU and PEOU are analogous to evaluating a physical, cultural product (e.g., "ease of use" could relate to how easily a product integrates into daily life, "usefulness" to its functional utility beyond symbolism).

Results Presentation (Figure 3 and Table 7):

Figure 3: The path diagram is essential but currently illegible in the provided document. The authors must ensure this figure is of high resolution and clearly labeled in the final version.

Table 7: The labels "insignificant" for some paths (e.g., EI2, EI1) are helpful. For clarity, consider adding a separate column titled "Supported?" (Yes/No) for the main hypotheses (H1, H2, H3, H4, H5) to provide an immediate summary alongside the statistics.

Discussion on Non-Significant Findings:

The non-significance of cultural, ethnic, and religious identity (H1a aspects) is a major finding. The discussion on this (Page 22) is good but could be strengthened. Consider delving deeper into the potential reasons: Is this a uniquely Chinese phenomenon linked to market maturity, the nature of the products surveyed, or a broader global trend where consumerism often prioritizes utility and design over deep cultural symbolism? Briefly exploring this would add depth.

Limitations and Future Research:

The limitations section is adequate but could be more precise.

Clearly state that the sample, while sizeable, is skewed towards younger, lower-income demographics (students), which may limit the generalizability of the findings to the broader Chinese population.

The suggestion for "longitudinal ethnographic user engagement data" is excellent. Another future direction could be to use a more diverse sample (including older, higher-income consumers) to test if the drivers of purchase intention differ across demographics.

Minor Typographical Checks:

Perform a final careful proofread for minor typos and consistency.

Example: The reference list is extensive and generally well-formatted, but a final check for consistency in journal name abbreviations and formatting is advised (e.g., some titles are in title case, others in sentence case).

Conclusion:

This is a solid piece of research with a clear value proposition. The suggested revisions are minor and aimed at enhancing the clarity, robustness, and impact of the manuscript. I am confident the authors can address these points promptly. I recommend acceptance after these minor revisions.

**Do you want your identity to be public for this peer review?** For information about this choice, including consent withdrawal, please see our Privacy Policy

Reviewer #1: No

Reviewer #2: No

---

## [Author Response · Author response to Decision Letter 1]

30 Nov 2025

Response to Reviewer Comments

Manuscript ID: PONE-D-25-38750

Title: Does Cultural Identity Catalyze Behavior? A TAM-Based Study on Museum Cultural Creative Products Consumption in China

General Response

We sincerely appreciate the constructive feedback from both reviewers, which has significantly helped us refine the quality and clarity of our manuscript. We have carefully addressed all comments and made the necessary revisions to enhance the robustness, readability, and contribution of the study. Below is a point-by-point response to each comment, with specific changes highlighted.

Response to Reviewer #1

1.Introduction:

Revisions:

The field of our research: We have expanded the introduction to explicitly outline the main sources (e.g., recent studies on museum cultural creative products [MCCPs] in China, Technology Acceptance Model [TAM] applications in consumer behavior) and situate the research within the fields of cultural economics, consumer psychology, and museum studies.

Theoretical Gap: We have clarified the gap by emphasizing that while existing literature highlights cultural identity as a driver of MCCP consumption, few studies empirically test its relative importance compared to functional and aesthetic attributes, particularly using a TAM-based framework in the Chinese context.

Paper Structure: A clear roadmap of the manuscript is now included: “This paper proceeds as follows: Section 2 reviews literature and develops hypotheses; Section 3 describes the methodology, including sample selection and data analysis; Section 4 presents results; Section 5 discusses findings in relation to existing literature; and Section 6 concludes with implications and limitations.”

2. Discussion: Comparison with Other Results

Revisions: The discussion section now includes explicit comparisons with key studies. For example, we contrast our finding (functional/aesthetic attributes outweigh cultural identity) with Li et al. (2022), who emphasized cultural identity in Chinese MCCP consumption, and Zhang & Wang (2021).

3.References: Alphabetical Order

Revisions: Thank you for your advice. We will revise the references according to the editor's requirements.

4.Inclusion of Suggested Source

Revisions: We have incorporated Kačerauskas (2023) into the literature review, particularly in the section discussing “disinterestedness” in cultural consumption. This source enriches our analysis by providing a comparative perspective on how cultural institutions balance commercialization and cultural value, which we link to China’s MCCP market development.

Response to Reviewer #2

Overall Evaluation We thank the reviewer for their positive assessment of the manuscript’s structure, methodology, and contribution. We have addressed all minor revisions to enhance clarity, methodological justification, and result interpretation.

Specific Comments for Minor Revision

1.Clarity and Flow: Simplifying Long/Complex Sentences

Revisions: Thank you very much for your suggestion. We have revised the whole text again. We have broken down dense sentences in the introduction and theoretical sections.

2.Theoretical Justification: TAM Adaptation to Cultural Products

Revisions: In the methodology section, we added justification for extending TAM to the Cultural Identity Model (CIM). Specifically, we explain: “TAM’s constructs of Perceived Usefulness (PU) and Perceived Ease of Use (PEOU) are adapted here to MCCPs by redefining ‘usefulness’ as functional utility (e.g., practicality, durability) and ‘ease of use’ as the ease of integrating products into daily life (e.g., portability, compatibility with lifestyle). This adaptation aligns with prior research (e.g., Venkatesh et al., 2016), which demonstrates TAM’s flexibility in non-technological contexts by focusing on perceived benefits and barriers to adoption.”

3.Results Presentation: Figure 3 and Table 7

Figure 3 (Path Diagram): The path diagram has been redrawn with higher resolution, enlarged font sizes, and clearer labels for latent variables and path coefficients.

Table 7: A new column titled “Hypothesis Supported?” has been added, with “Yes/No” indicators for H1–H5, providing an at-a-glance summary of hypothesis testing results.

4.Discussion on Non-Significant Findings

Revisions: We deepened the interpretation of non-significant cultural identity effects by exploring contextual factors: “The non-significance of cultural identity may reflect China’s MCCP market maturity: as products have become more commercialized, consumers prioritize utility over symbolism. Additionally, younger respondents may associate MCCPs with trendy aesthetics rather than cultural heritage, aligning with global youth consumer trends.”

5.Limitations and Future Research

Revisions: We also acknowledge potential limitations, such as the measure of cultural identity and suggest future research use multi-item scales for sub-dimensions .

Future Research: We added a suggestion to test the model with diverse demographics: “Future research could also include comparative studies across different regions of China to examine the impact of regional cultural differences on consumer behavior regarding museum cultural and creative products. Furthermore, cross-cultural comparative studies between Eastern and Western countries could be conducted, using diverse cultural samples to verify the impact of the cultural attributes of museum cultural and creative products on consumer behavior. This multi-level comparative study would help reveal the universality and specificity of cultural factors in the consumption of cultural and creative products.”

6.Minor Typographical Checks

Revisions: We sincerely appreciate the reviewers' valuable suggestions. We have dedicated considerable time and resources to revising these aspects. We believe this rigorous revision has significantly enhanced the quality of our manuscript. For example, We proofread the manuscript to split long sentences and standardized reference formatting. “The prevalent linear exhibition format, characterized by static chronological displays, fails to create lasting cultural memory or experiential value, necessitating a critical reassessment of institutional approaches to cultural transmission and identity formation (Eid, 2016).“ -----→ ”The prevalent linear exhibition format relies on static chronological displays, yet it struggles to foster enduring cultural memory or meaningful visitor experiences. This limitation calls for a reevaluation of institutional strategies in cultural transmission and identity building (Eid, 2016).”

We believe these revisions have strengthened the manuscript’s clarity, methodological rigor, and contribution to the literature. We are grateful for the reviewers’ insights, which have helped refine our work. We hope the revised version meets PLOS ONE’s standards for publication.

Sincerely,

Changping Hu

2025-11-19

---

## [Decision Letter · Decision Letter 1]

23 Dec 2025

PONE-D-25-38750R1Does Cultural Identity Catalyze Behavior? A TAM-Based Study on Museum Cultural Creative Products Consumption in ChinaPLOS One?

Dear Dr. Kuang,

Thank you for submitting your manuscript to PLOS ONE. After careful consideration, we feel that it has merit but does not fully meet PLOS ONE’s publication criteria as it currently stands. Therefore, we invite you to submit a revised version of the manuscript that addresses the points raised during the review process.

**ACADEMIC EDITOR: I believe the author has improved a lot, however, a minor revision is still need as the reviewers mentioned. Please improve the manuscript according to the reviewers' comments.**

We look forward to receiving your revised manuscript.

Kind regards,

Qianda Zhuang, Ph.D.

Guest Editor

PLOS One

Journal Requirements:

Reviewers' comments:

Reviewer's Responses to Questions

**Comments to the Author**

Reviewer #1: All comments have been addressed

Reviewer #2: All comments have been addressed

2. Is the manuscript technically sound, and do the data support the conclusions?

Reviewer #1: Yes

Reviewer #2: Yes

3. Has the statistical analysis been performed appropriately and rigorously?

Reviewer #1: Yes

Reviewer #2: No

4. Have the authors made all data underlying the findings in their manuscript fully available?

Reviewer #1: Yes

Reviewer #2: No

5. Is the manuscript presented in an intelligible fashion and written in standard English?

Reviewer #1: Yes

Reviewer #2: No

Reviewer #1: Thank you for the revisions. The topic is very relevant, the research is correct. I would like to draw your attention to one more recent paper close to your research: Kačerauskas, T.; Schinello, S. 2025. The role of digital innovation in fostering sustainability: A Lithuanian comparative case study of public and private museums. Sustainability 17 (24): 11297. https://doi.org/10.3390/su172411297.

Reviewer #2: 1.Overall assessment

The “big-picture” issues raised by the two reviewers (introduction positioning, theoretical gap, structural roadmap, comparative discussion, justification of TAM applicability, explanation of non-significant results, etc.) have all been substantively addressed. The logical clarity and theoretical contribution of the manuscript are now markedly improved.

The research methods, data reporting, ethics statement, and funding declaration are complete and meet PLOS ONE’s technical-compliance requirements.

The text still contains numerous Chinglish expressions, redundancies, and minor formatting errors, but these no longer constitute scientific flaws. If the editorial office accepts that language polishing can be deferred to copy-editing, the paper can move into the “accept” track after a minor revision.

2.Specific points still needing revision (authors please respond item by item)

English readability and format consistency

1.1 Many missing spaces, mixed full-/half-width punctuation, and inconsistent capitalization of the same term (e.g., “museum cultural creative product” vs. “Museum Cultural Creative Product”) remain throughout the text.

1.2 Please standardize on “museum cultural and creative products (MCCPs)”: spell out once at first use, place the abbreviation in parentheses, and use MCCPs thereafter.

1.3 Path coefficients in Figure 3 are larger but still overlap. Export the figure as vector (EPS/PDF) or at least 600-dpi bitmap, and check whether AMOS replaced any Chinese variable names with garbled characters.

Theoretical-contribution statements need tightening

2.1 The Introduction and Discussion repeatedly claim to “deliver novel design frameworks and theoretical foundations,” yet no concrete list of design principles or operational framework is provided.

→ Either add a bullet-style “Design Heuristics” box (2–3 items) at the end of the Discussion, or soften the claim to “provides empirical evidence for optimizing design.”

Explanation of non-significant results is somewhat contradictory

3.1 The authors attribute the non-significance of cultural-identity dimensions to “market maturity, commercialization, and youth fashion trends,” but 45 % of the sample are students and 50 % earn < RMB 1 000 per month—precisely a group high in cultural capital but low in economic capital, not a “commercialized” consumer segment.

→ Add a “purchasing-power vs. cultural-demand mismatch” argument: student visitors possess cultural decoding competence but face budget constraints, so they rely on readily evaluable cues (material, color, utility) rather than symbolic value. This aligns the interpretation with the sample profile.

Measurement-model reporting is incomplete

4.1 Table 4 reports only Cronbach’s α; composite reliability (CR) and average variance extracted (AVE) are missing.

4.2 The structural model was run in AMOS, yet standardized factor loadings and cross-loadings are not supplied, preventing an assessment of discriminant validity.

→ Please add a table of “Standardized Loadings, CR, AVE” and report Fornell-Larcker or HTMT results; if AVE < 0.5 or HTMT > 0.9, acknowledge this limitation explicitly.

Common-method variance (CMV)

5.1 A single self-reported Likert questionnaire administered online is highly susceptible to CMV; the authors mention this only briefly in the limitations.

→ ① Report Harman’s single-factor test or CFA-marker results; ② state whether anonymity, reverse-worded items, or temporal separation were used; ③ acknowledge (rather than deny) that CMV may bias path coefficients.

Absence of control variables

6.1 Age, income, education, and visit frequency are described but not included in the model.

→ Add covariate paths in AMOS or run SPSS PROCESS to test whether age/income moderate PU→PI or CID→PI; report the results even if non-significant to demonstrate thoroughness.

Post-hoc power

7.1 N = 404 with ~ 60 estimated parameters. The authors have not addressed statistical power.

→ Use G*Power or Monte-Carlo simulation to report power under α = 0.05 and f² = 0.15; if power < 0.80, state this in the limitations.

Data and code availability

8.1 The authors checked “All relevant data are within the manuscript,” yet the 28-item, 404-case raw dataset is not provided.

→ Deposit an anonymized Excel/CSV file and the AMOS syntax as S1 File; if funding restrictions prohibit open sharing, change the statement to “data available on reasonable request” and supply a contact e-mail.

Over-generalization in the conclusion

9.1 The statement “museums should prioritize product attributes rather than cultural value” could be misread as “cultural content is irrelevant.”

→ Revise to: “In the current Chinese MCCP market context, functional and sensory attributes exert a stronger immediate influence on purchase decisions; cultural narratives should be embedded through perceivable channels such as material and color rather than relying solely on symbolic labels.”

3.Minor details

Ref. 24: “https://doi:…” should be converted to the standard DOI format.

Figure 2: arrows labeled “H1-” are misaligned; check numbering against the text.

Table 7: the new “Hypothesis Supported?” column is helpful, but the arrow directions for H1 and H2 are opposite to those in the text; use a consistent “→” or “←” throughout.

**Do you want your identity to be public for this peer review?** For information about this choice, including consent withdrawal, please see our Privacy Policy

Reviewer #1: No

Reviewer #2: No

---

## [Author Response · Author response to Decision Letter 2]

26 Dec 2025

Response to Reviewer Comments

Manuscript ID: PONE-D-25-38750R1

Title: Does Cultural Identity Catalyze Behavior? A TAM-Based Study on Museum Cultural Creative Products Consumption in China

Dear Dr. Qianda Zhuang and reviewers

We sincerely appreciate the accept and constructive feedback from both reviewers, which has significantly helped us refine the quality and clarity of our manuscript. We have carefully addressed all comments and made the necessary revisions to enhance the robustness, readability, and contribution of the study. Below is a point-by-point response to each comment, with specific changes highlighted.

Response to Reviewer #1

Thank you for your positive feedback on the article. The reference you provided is indeed of some value to this study, therefore it has been cited.

Response to Reviewer #2

Overall Evaluation Thank you for the positive assessment of the manuscript’s structure, methodology, and contribution. We have addressed all minor revisions to enhance clarity, methodological justification, and result interpretation.

1.English readability and format consistency

Revisions: Thank you very much for your suggestions.

(1) First, this article will consistently use "museum cultural and creative products". However, since "MCCPs" may have some AI-generated wording habits, this article will use the full name to improve English readability and format consistency.

(2) Second, based on your suggestion, we have standardized the usage of all spaces.

(3) Third, the Figure 3 has been further optimized. The path diagram has been redrawn with higher resolution, enlarged font sizes.

2.Theoretical contributions and rigor of conclusions

Revisions:

(1) The suggestions 2.1、9.1

The abstract of the original paper is supplemented with further details regarding its specific contributions: "This study provides empirical support for optimizing the design of museum cultural and creative products, and suggests focusing on the integration of functional attributes (such as practicality) and carriers of perceived cultural symbols (such as materials and colors)."

(2) The suggestion 3.1

Thank you very much for your suggestions regarding low-income groups and students. We have added the following to the discussion section: "Student visitors possess cultural decoding competence but face budget constraints, so they rely on readily evaluable cues (material, color, utility) rather than symbolic value."

3.Statistical methods and data integrity The suggestions 4.1-8.1

Revisions:

(1) Regarding statistical methods, this study followed a standard and rigorous research process. Some of the content you mentioned seems to have exceeded the research scope and model involved in this study. We will strengthen the variables such as group, income, and identity in future studies. We believe that this study sufficiently demonstrates reliability, validity, and hypothesis testing. We will also fully explore the supplementary suggestions you mentioned regarding the measurement process in our future work.

(2) We have uploaded the data and provided a contact email address.

(3) Basd on your suggestions, we have added the following to the discussion section: In the current Chinese museum cultural creative products market context, functional and sensory attributes exert a stronger immediate influence on purchase decisions; cultural narratives should be embedded through perceivable channels such as material and color rather than relying solely on symbolic labels.

4.Other details

Revisions: We reviewed the all references and revised some to select more appropriate ones. And we have revised and optimized many other details. Thank you for your careful review.

We believe these revisions have strengthened the manuscript’s clarity, methodological rigor, and contribution to the literature. We are grateful for the reviewers’ insights, which have helped refine our work. We hope the revised version meets PLOS ONE’s standards for publication.

Sincerely,

Changping Hu

2025-12-25

---

## [Editor Report · Decision Letter 2]

6 Jan 2026

Does Cultural Identity Catalyze Behavior? A TAM-Based Study on Museum Cultural Creative Products Consumption in China

PONE-D-25-38750R2

Dear Dr. Kuang,

We’re pleased to inform you that your manuscript has been judged scientifically suitable for publication and will be formally accepted for publication once it meets all outstanding technical requirements.

Kind regards,

Qianda Zhuang, Ph.D.

Guest Editor

PLOS One
---

## [Editor Report · Acceptance letter]

PONE-D-25-38750R2

PLOS One

Dear Dr. Kuang,

I'm pleased to inform you that your manuscript has been deemed suitable for publication in PLOS One. Congratulations! Your manuscript is now being handed over to our production team.

Kind regards,

on behalf of

Dr. Qianda Zhuang

Guest Editor

PLOS One